# DABCO-Catalyzed Mono-/Diallylation of *N*-Unsubstituted Isatin *N*,*N*′-Cyclic Azomethine Imine 1,3-Dipoles with Morita-Baylis-Hillman Carbonates

**DOI:** 10.3390/molecules28073002

**Published:** 2023-03-28

**Authors:** Qiumi Wang, Sicheng Li, Guosheng Yang, Xinyu Zou, Xi Yin, Juhua Feng, Huabao Chen, Chunping Yang, Li Zhang, Cuifen Lu, Guizhou Yue

**Affiliations:** 1College of Science, Sichuan Agricultural University, Ya’an 625014, China; 2The Yingjing County Emergency Management Agency, Ya’an 625200, China; 3College of Agronomy, Sichuan Agricultural University, Chengdu 611130, China; 4Hubei Collaborative Innovation Center for Advanced Organochemical Materials, Ministry-of-Education Key Laboratory for the Synthesis and Application of Organic Functional Molecules, Hubei University, Wuhan 430062, China

**Keywords:** azomethine imine, allylation, dipole, MBH carbonate, cycloaddition

## Abstract

Allylation of *N*-unsubstituted isatin *N*,*N*′-cyclic azomethine imines with Morita-Baylis-Hillman carbonates in the presence of 1–10 mol% DABCO in DCM at room temperature, rapidly gave *N*-allylated and *N*, *β*-diallylated isatin *N*,*N*′-cyclic azomethine imine 1,3-dipoles in moderate to high yields. The reaction features mild reaction conditions, easily practical operation, and short reaction times in most cases. Furthermore, the alkylated products were transformed into novel bicyclic spiropyrrolidine oxoindole derivatives through the [3+2] or [3+3]-cycloaddition with maleimides or Knoevenagel adducts.

## 1. Introduction

Heterocycles are privileged structural units that are frequently encountered in biologically active natural products as well as in pharmaceuticals and agrochemicals [1,2,3]. In particular, the pyrazole-ring skeletons, including pyrazolone and pyrazolidinone, are the core skeletons in many biologically active compounds. For example, edaravone (I) was used as a free radical scavenger for the treatment of amyotrophic lateral sclerosis (ALS) (Figure 1) [4]. Antipyrine (II) showed analgesic and antipyretic activities [5], and the analogue aminophenazone (III) also demonstrated antipyretic and anti-inflammatory activities [6]. In addition, Metamizole (IV) has been considered the strongest antipyretic drug for perioperative and cancer pain [7]. Eltrombopag (V) was used for the treatment of low blood platelet counts in adults with idiopathic chronic immune thrombocytopenia [8]. Furthermore, sulfamazone (VI) was regarded as a drug candidate for anti-inflammatory activity [9]. Therefore, the exploration of practical and efficient methods for the synthesis of dinitrogen-fused heterocycles has attracted extensive attention in the field of organic chemistry and pharmacology. Moreover, 3-substituted oxindoles are also heterocyclic frameworks and have widely existed in bioactive molecules [10,11,12,13,14,15,16]. Various methods of constructing 3-substituted oxindoles have been reported [17,18,19].

Very recently, Jin and co-workers [20] envisioned a direct pathway to access 3,3-spiropiperidine oxindoles via the [3+3]-annulation of the isatin *N*,*N*′-cyclic azomethine imine 1,3-dipole with Morita-Baylis-Hillman (MBH) carbonates in tertiary amines or phosphines. However, the authors did not find the above products but observed the allylation of isatin *N*,*N*′-cyclic azomethine imines with MBH carbonates catalyzed by 4-dimethylaminopyridine (DMAP), which generated 16 corresponding products in excellent yields (78–99%) (Figure 1). There was a disadvantage when *N*-unsubstituted isatin *N*,*N*′-cyclic azomethine imine was employed as a substrate, as the reaction only afforded a trace amount of allylated product. To date, the reactions of isatin *N*,*N*′-cyclic azomethine imines were rarely studied and demonstrated by only a few examples [21,22,23,24,25,26,27,28,29,30,31,32,33,34,35,36]. Therefore, it is urgent to explore the new reaction of isatin *N*,*N*′-cyclic azomethine imines. MBH adducts contain the structural moieties of allylic alcohols or amines, Michael acceptors, and electron-withdrawing groups, which makes them valuable substrates for various types of reactions such as Michael addition, allylic substitution, cycloaddition reaction, Friedel-Crafts reaction, Claisen rearrangement, etc. [37,38,39,40,41,42,43,44,45,46,47,48,49,50,51,52,53,54,55,56,57,58]. Based on our previous studies of 1,3-dipolar cycloaddition, Michael addition of azomethine ylides and azomethine imines and palladium-catalyzed tandem reaction to construct 3,3-disubstituted indolinones [59,60,61,62,63,64,65,66,67,68], herein we report the mono-/diallylation of isatin *N*,*N*′-cyclic azomethine imines from the condensation of isatin and pyrazolidones with MBH carbonates.

## 2. Results and Discussion

Before starting this work, we found that *N*-unprotected isatin *N*,*N*′-cyclic azomethine imine **1a** reacted with MBH carbonate **2a** in the presence of 5 mol% DMAP in dichloromethane (DCM) at room temperature in 45 min via *N*- and C_β_-allylation; this gave the corresponding **3a** and **4a**, respectively, in 24% and 17% yields (Table 1, entry 1). Jin’s group [20] reported that **1a** reacted with **2a**, only to obtain a trace amount of *β*-allylated product in the presence of 20 mol% DMAP in DCM at refluxing (**3a** or **4a** was not observed). The above result encouraged us to explore and develop *N*-and *β*-allylation as a supplement to their approach.

Subsequently, we found that DABCO, instead of DMAP, quickly gave *N*-allylated product **3a** with a satisfactory yield in the same condition (entry 2). To improve the yield and regioselectivity of the reaction, the reaction conditions were optimized. First, the solvents were investigated. In the chloroalkanes, both chloroform and dichloroethane (DCE), the reactions led to inferior results in contrast with DCM (entries 3 and 4). The aprotic polar solvents, dimethylsulfoxide (DMSO), *N*,*N*-dimethylformamide (DMF) and *N*,*N*-dimethylacetamide (DMA), also gave poor yields and regioselectivities. Other solvents, such as ethyl acetate, acetonitrile (ACN), and ethers (for example, diethyl ether, tetrahydrofuran (THF), and dioxane), led to unsatisfying results. Therefore, DCM was selected as the best solvent. Second, various organic and inorganic bases were screened. When common tertiary amines were used, including triethylamine (TEA), diisopropylethylamine (DIPEA), and 1,8-diazabicyclo[5.4.0]undec-7-ene (DBU), the yield of the *N*-allylated product was lower than using DMAP. In inorganic bases, NaOH, KOH, and NaH, only 7–13% yields were obtained, while in Na_2_CO_3_, K_2_CO_3_, and Cs_2_CO_3_, the reaction did not work at all. Triphenylphosphine made the reaction yield diallylated product **4a** with poor yield (26%). Combined with the above results, DABCO was selected as the base. Next, the loading amounts of DABCO were screened. When a 1 mol% loading amount was used, the yield was better than the 5 mol% loading amounts (entry 24), in which 10 mol% loading amounts conversely gave an inferior yield (entry 25). In addition, the concentration of the reaction and equivalent of MBH carbonate **2a** were also screened to find that the reaction gave the best yield (91%) in the presence of 2.2 equivalent **2a**. When the reaction time was extended to 7 h and 10 mol% DABCO was used, only the diallylated product **4a** was formed in 77% yield. The optimal reaction condition for monoallylation was established, and the desired product could be obtained in 91% yield when using isatin *N*,*N*′-cyclic azomethine imine **1a** (1 equiv.), MBH carbonate **2a** (2.2 equiv.), and catalyst DABCO (1 mmol%) in DCM at rt for 30 min (entry 28). The optimal reaction condition for diallylation afforded 77% of the product yield when using isatin *N*,*N*′-cyclic azomethine imine **1a** (1 equiv.), MBH carbonate **2a** (2.2 equiv.), and catalyst DABCO (10 mmol%) in DCM at rt for 7 h (entry 33).

After establishing the optimal reaction conditions, a wide range of different substituted aryl isatin *N*,*N*′-cyclic azomethine imines have been explored for this nucleophilic substitution reaction. As summarized in Table 2, various substituent groups employed on the isatin moiety of **1** could be tolerated which afforded the desired products with moderate to excellent yields (49–91%) (Table 2, entries 1–8), except for 5-nitro isatin *N*,*N*′-cyclic azomethine imine **1d** (entry 9). The reaction of **1a** with **2a** under a 1 mmol scale with the same yield (91%) compared with under 0.5 mmol at the most optimal conditions. It is worth noting that the corresponding products **3a** could be afforded in 84% yield (1.28 g) when istain *N*,*N*′-cyclic azomethine imine **1a** was scaled up to 4.65 mmol. The substituent patterns on the benzene ring of azomethine imines had a vital impact on the yields. Overall, the yields dropped off, whether it is electron-withdrawing or electron-donating groups, particularly the 7-CF_3_ group (entry 9). To our surprise, 5-nitro isatin *N*,*N*′-cyclic azomethine imine reacted with **2a**, to give *C3*- and *N*-diallylated product **3′i**, but not **3i** within a short time (1 min) (Figure 2). The structure of **3′i** was confirmed unambiguously by single-crystal X-ray diffraction [69]. Various MBH carbonates (R^1^ = Me, *n*-Pr, *n*-Bu, and *t*-Bu) also reacted smoothly, in which the yields were 40–80%.

Subsequently, the generality of the allylation was further demonstrated using various aryl MBH carbonates. As outlined in Table 3, it is regrettable that all the yields of examples were not better than that of the model reaction, regardless of electron-donating groups and electron-withdrawing groups in phenyl. All the results showed that these reactions gave a complex when the aryl groups of MBH carbonates were 4-MeOC_6_H_4_, 4-FC_6_H_4_, 2-BrC_6_H_4_, and 2-NO_2_C_6_H_4_ (entries 7, 10, 14, and 17). These reactions led to the desired products in low yields or with an inseparable by-product (see Appendix A) when aryl groups were 2-MeC_6_H_4_, 2-MeOC_6_H_4_, 2-FC_6_H_4_, and 2-ClC_6_H_4_ (entries 2, 5, 8, and 11). To our surprise, 3-thiophenyl MBH carbonate also afforded the diallylated product **6′t** (Figure 3), similarly to that of isatin *N*,*N*′-cyclic azomethine imine bearing a 5-NO_2_ group in benzene ring (Figure 3).

Various *N*-substituted isatin *N*,*N*′-cyclic azomethine imines **8** (R = alkyl, allyl, Bn, and propargyl) that were not used for testing by Jin’s group except for **8c**, could also react with MBH-carbonate **2a** with moderate to excellent yields (Table 4, entries 2–11) in our optimal condition. It is surprising that *N*-methyl isatin *N*,*N*′-cyclic azomethine imine **7a** hardly reacted with **2a** in the standard condition (entry 1). Among them, *N*-Bn isatin *N*,*N*′-cyclic azomethine imine could offer the desired product with a good yield (75%), though not as high as the yield (92%) reported by Jin’s group (entry 3). The reaction of *N*-allyl isatin *N*,*N*′-cyclic azomethine imine **7d** gave the best result (82% yield), using **2a** as a partner (entry 4). However, *N*-propargyl isatin *N*,*N*′-cyclic azomethine imine only gave a 20% yield, because of some side reactions (entry 5).

To expand the application of the reaction, isatin *N*,*N*′-cyclic azomethine imines **1** reacted with MBH carbonates **2** in the presence of 10 mol% DABCO and prolonged the reaction time, which afforded diallylated products **4** in 41–77% yield (Table 5). On the whole, the yields of all reactions were not high, except for **4e** and **4i**. The possible reason is that the prolonged reaction time leads to increasing side reactions.

The reaction of α-methyl isatin *N*,*N*′-cyclic azomethine imine **9** with MBH carbonate **2a** was tested, which successfully obtained a corresponding product **10** in excellent yield (84%) within 2 min (Figure 4). Meanwhile, *β*-phenyl isatin *N*,*N*′-cyclic azomethine imines **11** could also obtain the desired product **12** with a satisfied yield (77%) within 10 min.

The *N*-allylated product **3a** exhibited a potentially wide application in organic synthesis (Figure 5). For example, the Michael addition of **3a** with *β*-nitrostyrene in the presence of DABCO provided 3,3-disubstituted oxindole **13** in 37% yield with >20:1 *dr*, while no product was obtained in the condition reported by Wang’s group [22]. The [3+3] cycloaddition of **3a** with Knoevenagel adduct under K_2_CO_3_/DCE could afford spiropyridazine oxoindole **14** in 60% yield with >20:1 *dr*, which the [3+2] cycloaddition of **3a** with maleimide also produced tricyclic spiropyrrolidine oxoindole **15** in 91% yield with >20:1 *dr*. Finally, **3a** could be converted into diallylated product **4a** with moderate yield (64%).

Based on the literature reports [20], our results, and X-ray analysis, a plausible mechanism is proposed for the formation of **3a**, **3′I,** and **4a** (Figure 6). First, isatin *N*,*N*′-cyclic azomethine imine **1** reacted with MBH carbonate **2** in the presence of DABCO, to obtain *N*-alkylated products **3** or **6**. The resonance form **In-A** of **3** or **6** quickly tautomerized to the delocalized intermediate **In-B** under DABCO. Second, pyrazolenone intermediate **In-C** could be generated from **In-B**, then promote the isatin carbanion to react with MBH carbonate **2** through *β*-allylation, with the corresponding product **4** obtained. Moreover, when R was a nitro group, the delocalized intermediate **In-B** preferred to proceed with *C3*-allylation, followed by a Boc-protected reaction of the hydroxy group, to achieve *N*- and *C3*-diallylated product **3′i**.

## 3. Materials and Methods

### 3.1. General Methods

All reactions were carried out without strict water-free and oxygen-free conditions. All solvents and reagents were obtained from commercial suppliers and were directly used for reactions without further purification unless otherwise stated. When the reactions were performed at the condition of NaH, DCM was pre-dried with CaH_2_. Flash chromatography was performed using silica gel (200–300 mesh). Reactions were monitored by TLC or/and colour changes of the reaction solution. Visualization was achieved under a UV lamp (254 nm and 365 nm), I_2_, and by developing the plates with phosphomolybdic acid. ^1^H and ^13^C NMR were recorded on 400 and 600 MHz NMR spectrometers with tetramethylsilane (TMS) as the internal standard. The chemical shift values were corrected to 7.26 ppm (^1^H NMR) and 77.16 ppm (^13^C NMR) for CDCl_3_. IR spectra were acquired on an FT-IR spectrometer and are reported in wavenumbers (cm^−1^). High-resolution mass spectra were obtained using electrospray ionization (ESI). ^1^H NMR splitting patterns are designated as singlet (s), double (d), broad singlet (br s), triplet (t), quartet (q), doublet of doublets (dd), multiples (m), etc. Coupling constants (J) are reported in Hertz (Hz).

### 3.2. Preparation of Intermediates

Pyrazolidine-3-ones were obtained by the reaction of hydrazone monohydrate with methyl acrylate in ethanol under refluxing conditions [21]. All isatin *N*,*N*′-cyclic azomethine imines **1** were prepared by the condensation of isatins and the above pyrazolidone in menthol under 45 °C or a refluxing condition [21]. All MBH carbonates **2** were prepared by two-step reactions, including the Morta–Maylis–Hillman reaction (1 equiv. DABCO/1 equiv. aldehyde/1.5 equiv. acrylate/1:1 dioxane:H_2_O or THF/2–3 days) [70] and the formation of an O-Boc derivative (0.1 equiv. DMAP/1 equiv. MBH alcohol/1.5 equiv.Boc_2_O/DCM/rt/overnight), with 22–64% total yields [71].

### 3.3. General Procedure for Condition Optimization

A 10 mL tube was charged with isatin *N*,*N*′-cyclic azomethine imine **1a** (0.5 mmol, 1.0 equiv.), MBH carbonate **2a** (0.55–1.65 mmol, 1.1–3.3 equiv.), base (0.005–0.1 mmol, 1–20 mol%), and solvent (1.0–4.0 mL). The suspended solution was vigorously stirred at rt-reflux, and then the base was added. When the suspension reaction liquid became completely clear, the reaction had finished. The solution was added by 5 mL H_2_O and 15 mL brine before the resulting mixture was extracted with DCM (5 × 10 mL). The combined organic layers were dry with Na_2_SO_4_, filtered, and concentrated. The residue was purified by flash silica gel chromatography eluated with EtOAc:PE (1:5 to 1:1) to afford the corresponding products **3a** and/or **4a**.

### 3.4. General Procedure for Typical Procedure for Monoallylation

A tube (25 mL) was charged with isatin *N*,*N*′-cyclic azomethine imine **1a** (1.0 mmol, 1.0 equiv.), MBH carbonate **2a** (2.2 mmol, 2.2 equiv.), and DCM (4 mL). The suspended solution was vigorously stirred at rt, and then DABCO (0.01 mmol, 0.01 equiv., 1 mol%) was added. When the reaction mixture became clear the reaction finished (2–30 min). The solution was added by 10 mL H_2_O and 30 mL brine before the resulting mixture was extracted with DCM (5 × 20 mL). The combined organic layers were dried with Na_2_SO_4_, filtered, and concentrated. The residue was purified by flash silica gel chromatography eluated with EtOAc:PE (1:5 to 1:1) to afford the corresponding monoallylated products **3**, **6**, **8**, **10**, and **12**.

### 3.5. General Procedure for Typical Procedure for Dialkylation

A tube (25 mL) was charged with isatin *N*,*N*′-cyclic azomethine imine **1a** (1.0 mmol, 1.0 equiv.), MBH carbonate **2a** (2.2 mmol, 2.2 equiv.), and DCM (4 mL). The suspended solution was vigorously stirred at rt, and then DABCO (0.1 mmol, 0.1 equiv., 10 mol%) was added. When the reaction mixture became clear the reaction finished (5–12 h). The solution was added by 10 mL H_2_O and 30 mL brine before the resulting mixture was extracted with DCM (5 × 20 mL). The combined organic layers were dried with Na_2_SO_4_, filtered, and concentrated. The residue was purified by flash silica gel chromatography eluated with EtOAc:PE (1:5 to 1:1) to afford the corresponding diallylated products **4**.

### 3.6. Deriverziation of ***3a***

DABCO (11 mg, 0.10 mmol, 0.2 equiv.) was added to a solution of **3a** (164 mg, 0.5 mmol, 1.0 equiv) and β-nitrostyrenes (224 mg, 1.5 mmol, 1.5 equiv) in CHCl_3_ (2.0 mL) at rt. The mixture was stirred at rt for 2.0 h. The resulting mixture was a saturated NH_4_Cl solution (10 mL). The aqueous solution was extracted with EtOAc (3 × 15 mL). The combined organic layers were dried over Na_2_SO_4_, filtered, and concentrated under reduced pressure. The residue was purified by flash silica gel chromatography eluated with petroleum ether:EtOAc (3:1 to 1:1) to furnish Michael adduct **13**.

K_2_CO_3_ (138 mg, 1 mmol, 2.0 equiv.) was added to a solution of **3a** (164 mg, 0.5 mmol, 1.0 equiv) and 2-benzylidenemalononitrile (85 mg, 0.55 mmol, 1.1 equiv) in DCE (2.0 mL) at rt. The mixture was stirred at 83 °C for 20 min. The resulting mixture was a saturated NH_4_Cl solution (10 mL). The aqueous solution was extracted with DCM (3 × 15 mL). The combined organic layers were dried over Na_2_SO_4_, filtered, and concentrated under reduced pressure. The residue was purified by flash silica gel chromatography eluated with petroleum ether:EtOAc (5:1 to 1:1) to furnish cycloadduct **14**.

DABCO (11 mg, 0.10 mmol, 0.2 equiv.) was added to a solution of **3a** (164 mg, 0.5 mmol, 1.0 equiv) and maleimide (97 mg, 1.0 mmol, 2.0 equiv) in CHCl_3_ (2.0 mL) at rt. The mixture was stirred at rt for 2.0 h. The resulting mixture was a saturated NH_4_Cl solution (10 mL). The aqueous solution was extracted with EtOAc (3 × 15 mL). The combined organic layers were dried over Na_2_SO_4_, filtered, and concentrated under reduced pressure. The residue was purified by flash silica gel chromatography eluated with petroleum ether:EtOAc (3:1 to 1:1) to furnish cycloadduct **15**.

A tube (25 mL) was charged with **3a** (1.0 mmol, 1.0 equiv.), MBH carbonate **2a** (2.2 mmol, 2.2 equiv.), and DCM (4 mL). The suspended solution was vigorously stirred at rt, and then DABCO (0.1 mmol, 0.1 equiv., 10 mol%) was added. When the reaction mixture became clear the reaction finished (5–12 h). The solution was added by 10 mL H_2_O and 30 mL brine before the resulting mixture was extracted with DCM (5 × 20 mL). The combined organic layers were dried with Na_2_SO_4_, filtered, and concentrated. The residue was purified by flash silica gel chromatography eluated with EtOAc:PE (1:5 to 1:1) to afford the corresponding allylated adduct **4**.

## 4. Conclusions

In summary, we have developed a general method of DABCO-catalyzed mono-/diallylation of isatin *N*,*N*′-cyclic azomethine imine 1,3-dipoles with MBH carbonates. Various mono and diallyl isatin *N*,*N*′-cyclic azomethine imines are afforded in moderate to excellent yields (21–91%). All the synthesized compounds **3**, **3′i**, **4**, **6**, **6′t**, **8**, **10**, **12**, **13**, **14**, and **15** were confirmed through ^1^H and ^13^C NMR, IR, and HMRS technologies (see Appendix A). Furthermore, product **3a** can be transformed into functionalized compounds by cycloaddition and Michael to demonstrate the synthetic utilities. Further exploration and application of this reaction in organic synthesis are ongoing in our laboratory.

## Data Availability

Not applicable.

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
