# Peer review of "DABCO-Catalyzed Mono-/Diallylation of N-Unsubstituted Isatin N,N′-Cyclic Azomethine Imine 1,3-Dipoles with Morita-Baylis-Hillman Carbonates"

_molecules, 2023, doi:10.3390/molecules28073002_

Round 1

Reviewer 1 Report

After reading the manuscript entitled "DABCO-catalyzed mono-/diallylation of N-unsubstituted isatin N,N'-cyclic azomethine imine 1,3-dipoles with Morita-Baylis-Hillman carbonates" by Qiumi Wang et al., (Manuscript No: molecules-2290215), I consider it as acceptable for publication but minor changes should be introduced.  The work is well organized although I had a problem to understand the meaning of some parts. To the large extent, the manuscript is complemental to that cited as the Ref. [20], and these two papers together provide a nice story about the reactivity of isatin N,N'-cyclic azomethine imine 1,3-dipoles with Morita-Baylis-Hillman carbonates.  

The main objection relates to their repetitive claim that the reaction was conducted without specific precautions regarding moisture and oxygen. That can be the reason for the moderate yields that rarely exceed 85 %. I would suggest that authors add at least two DABCO-catalyzed reactions conducted under an inert atmosphere in the dried solvent. The reactions given in Entry 2 (Table 1) and Entry 5 (Table 5) could be the candidates.

Related to this point I am curious why the yields of the same reactions (Entry 2, Table 1 and Entry 1, Table 2) are so different (76 % vs 91 %)?

My other objections and suggestions are merely related to the typos and readability of the text:

1. Scheme 1 ("This work" part of the scheme): R2 is missing on two structures of products.

2. I would suggest rephrasing the second part of the first sentence in the Results and discussion section starting with "..., which Jin's group ...".  The cited paper mentions that, when N-unsubstituted isatine-based dipole was used, the only trace of the product they were looking for was formed. This does not necessarily imply low reactivity.  Also, different conditions authors used should not be termed as a "new methodology" (the end of the same paragraph).

3. In several instances two words were merged together - that should be checked.

4. The paragraph above Table 2 (page 5): "... azomethine mines ..." should be "... azomethine imines ..."

5. Table 2. I am wondering why the information about the presence or absence of the diallylated products is not present. Is it possible that in some cases lower yields of monoallylated products are due to the more efficient diallylation?

6. The first paragraph on the 8th page: (table 4) should be (Table 5).

7. The paragraph just before the Materials and Methods section:  "... could be generated from, then promote the isatin carbanion..." Something is missing here; from what?

8. Sections 3.6 (TLC) and 3.8 (NMR and other data) should be moved to Suppl. Mat.

9. Some labels used within the proposed mechanism (Scheme 4) are the same as compounds shown in Figure 1. They should be unique within the manuscript.

Reviewer 2 Report

Based on the previous related studies, Yue and his co-authors give us an article titled “DABCO-catalyzed mono-/diallylation of N-unsubstituted isatin N,N'-cyclic azomethine imine 1,3-dipoles with Morita-BaylisHillman carbonates”. Many compounds were synthesized and characterized by FTIR, NMR, and mass spectral analysis, importantly, structure of an accidental product 3’i was confirmed by XRD analysis. However, some unsatisfied details must be revised, as a result, it is not suitable for publication at this moment.

(1) I am very puzzled why this manuscript is present as Molecules 2023, 28, 1034. https://doi.org/10.3390/molecules28031034.

(2) Both of the abstract and introduction should be rewritten with the aim of more concise and relevant with the title. In my opinion, the authors should not pay the attention to introduce the pyrazole-ring skeletons in the part of introduction.

(3) The serial numbers showing in tables, figures, schemes, and the text should match well, please check them carefully. For example, in page 6, “afforded diallylated product 6't (Scheme 3)”, here the Scheme 3 should be Scheme 2; in page 8, “a corresponding product 10 in excellent yield (84%) within 2 min (Scheme 4)”, here the Scheme 4 should be Scheme 3; “diallylated products 4 in 31-77% yield (table 4)”, here the table 4 should be table 5. In addition, the yield of which diallylated product is 31%?? In scheme 2, “The reaction of 5-nitro-N,N'-cyclic azomethine imine 1a with thiophenyl MortaBaylis-Hillman carbonate 5o”, here the thiophenyl MBH carbonate is 5o? right? It should be 5t.

(4) In page 5, “Various MBH carbonates (R= Me, n-Pr, n-Bu and tBu) also reacted smoothly, in which the yields were 40-80%.”, here R should be R1.

(5) Many tables, figures, and schemes used in this paper, and it looks like an experiment report. Thus, the authors should reorganize these results to be more logical.

Reviewer 3 Report

Yue and collaborates have reported the development of a general procedure for the DABCO-catalysed mono/dialkylation of isatin N, N’-cyclic azomethine imine 1,3-dipoles with Morita-Baylis-Hillman carbonates. The products were fully characterized through spectroscopic techniques (1H, 13C, and IR) and spectrometric analysis, deeply reported in the paper and in the supplementary material. Furthermore, the synthetic procedures were reported in detail. The paper represents an important tool for studying and the optimization of the alkylation reaction.

I suggest to the authors of the work to improve Scheme 2, and Figure 2 by reporting the structure of the reactants, to facilitate the comprehension of the reaction.

Once these changes are made, I believe the work can be published on Molecules.

Reviewer 4 Report

The manuscript describes the regioselective allylation of N,N'-cyclic azomethine imines with MBH carbonates, representing a novel mode of reactivity of these substrates. Thorough optimization of reaction conditions was carried out to elaborate a simple and efficient preparative approach to novel derivatives of azomethine imines. Scope and limitations of the method were investigated and further application of products in organic synthesis was demonstrated. The design of the investigation is reasonable, the results of the work are supported with a proper set of methods including X-Ray analysis.

The manuscript may be published in Molecules after the following corrections.

1.      Abstract should be rewritten according to the Journal rules. Experimental details should be omitted.

2.      The language needs to be seriously improved. Some mistakes are marked with colour in the attached file, though it is not exhaustive list.

3.      Taking into account the poor yields of compounds 3 and 4 in some cases, was the formation of a mixture of mono- and diallylation products observed? Authors should comment on this point.

4.      Description of 1H NMR spectra of 1,2-disubstituted aromatic rings as a set of doublets and triplets is not correct. Spin–spin coupling constants other than 3J always make the structure of spectra more complex, even for some reasons, probably, connected with not optimal parameters of NMR experiments, it is not observed.

5.      HRMS data of some compounds (e.g. 3a,3’i,3k,6m) have the difference of Calculated and Found values more than 0.003 m/z units, that is unsufficient to support the molecular formula assignment.

Round 2

Reviewer 2 Report

I recommend it for publication without further modification.